# Polyurethane-Nanolignin Composite Foam Coated with Propolis as a Platform for Wound Dressing: Synthesis and Characterization

**DOI:** 10.3390/polym13183191

**Published:** 2021-09-20

**Authors:** Zari Pahlevanneshan, Mohammadreza Deypour, Amirhosein Kefayat, Mohammad Rafienia, Paweł Sajkiewicz, Rasoul Esmaeely Neisiany, Mohammad Saeid Enayati

**Affiliations:** 1Department of Biomaterials, Tissue Engineering and Nanotechnology, School of Advanced Medical Technologies, Isfahan University of Medical Sciences, Isfahan 81746-73461, Iran; 2Polymer Chemistry Research Laboratory, Department of Chemistry, Isfahan 81746-73441, Iran; mohammad.deypour737@gmail.com; 3Cancer Prevention Research Center, Department of Oncology, Isfahan University of Medical Sciences, Isfahan 81746-73461, Iran; Ahkefayat@yahoo.com; 4Biosensor Research Center, Isfahan University of Medical Sciences, Isfahan 81746-73461, Iran; 5Institute of Fundamental Technological Research, Polish Academy of Sciences, Pawinskiego 5B, 02-106 Warsaw, Poland; psajk@ippt.pan.pl; 6Department of Materials and Polymer Engineering, Hakim Sabzevari University, Sabzevar 96179-76487, Iran

**Keywords:** polyurethane foam, nanolignin, propolis, wound dressing

## Abstract

This piece of research explores porous nanocomposite polyurethane (PU) foam synthesis, containing nanolignin (NL), coated with natural antimicrobial propolis for wound dressing. PU foam was synthesized using polyethylene glycol, glycerol, NL, and 1, 6-diisocyanato-hexane (NCO/OH ratio: 1.2) and water as blowing agent. The resultant foam was immersed in ethanolic extract of propolis (EEP). PU, NL-PU, and PU-NL/EEP foams were characterized from mechanical, morphological, and chemical perspectives. NL Incorporation into PU increased mechanical strength, while EEP coating showed lower strength than PU-NL/EEP. Morphological investigations confirmed an open-celled structure with a pore diameter of 150–200 μm, a density of nearly 0.2 g/cm^3,^, and porosity greater than 85%, which led to significantly high water absorption (267% for PU-NL/EEP). The hydrophilic nature of foams, measured by the contact angle, proved to be increased by NL addition and EEP coating. PU and PU-NL did not show important antibacterial features, while EEP coating resulted in a significant antibacterial efficiency. All foams revealed high biocompatibility toward L929 fibroblasts, with the highest cell viability and cell attachment for PU-NL/EEP. In vivo wound healing using Wistar rats’ full-thickness skin wound model confirmed that PU-NL/EEP exhibited an essentially higher wound healing efficacy compared with other foams. Hence, PU-NL/EEP foam could be a promising wound dressing candidate.

## 1. Introduction

Various wound dressings have been applied for decades, as a practical way to protect the wound site on the skin from the exterior impact and provide conditions for absorbing excessive exudates [1,2]. Among different types of natural and synthetic polymers that have been used for wound dressing fabrication, polyurethane (PU) has been widely applied as a cheap raw material for dressings [3,4]. PU is a versatile polymer with fascinating applications [5], which are generally produced by the reaction of polyol, isocyanate, and chain extender in different forms (e.g., foam, film, hydrogel, and hydrocolloid) according to the production method [6,7]. In particular, PU foam is fabricated by using a one-step polymerization reaction and foaming technique with the addition of a blowing agent, commonly water [8].

PU foam is extensively used as wound dressings in the market due to its good biocompatibility, suitable flexibility, softness, low cytotoxicity, and acceptable mechanical property even after fully water immersing and also more economical in comparison with other natural polymer dressing materials [9,10].

Furthermore, the open-pore structure [11] in porous polyurethane foam dressings gives excellent cellular ingrowth in interconnected pores, water absorption capability, and a high moisture vapor transmission rate, which makes them suitable for a moderate to high volume of exudate absorption and to create a moist environment around the wound, which would accelerate wound healing in dermal wounds and prevent the wound infection depending on the PU foam properties such as thickness, texture, and pore size [12]

Polyurethanes can blend with different fillers to produce PU composite foam with unique characteristics [13]. Among the various fillers, naturally occurring counterparts have gained more attention, considering strict environmental concerns. Meanwhile, lignin is the second most abundant natural biopolymer composed of aromatic units that contain a lot of methoxy and hydroxyl groups in their amorphous structure [14]. Lignin has a wide range of physical, chemical, and biological properties, which has been always viewed by researchers as a promising material and can be considered as a potential biomaterial for biomedical applications such as tissue engineering, pharmaceuticals, drug delivery, and wound dressings due to its anti-oxidant and anti-microbial activities. Furthermore, various functional groups in lignin such as phenolic, carbonyl, carboxyl, and aliphatic hydroxyls can be used as chemical cross-linking agent reaction sites and physical hydrogen bonds [15,16].

Taking into account particular features of materials in nano-scale, in the last decade, many research studies have highlighted the reforming of lignin to functional nanomaterials such as lignin nanoparticles, namely nanolignin (NL), which is sized from 1 to 1000 nm [17]. Such reforming will significantly improve properties of NL, such as antioxidant activity, due to their high surface-area-to-volume, when compared with parent lignin. Overall, the synthesis of NL is a simple and controllable procedure that resulted in uniform nanoparticles formation. Among various methods, ultrasonication has the advantages of being a green synthesis, fast, low cost, and high yield method [17].

Development of nanocomposites using NL, due to being abundantly and easily available, inexpensive, and biocompatible, for applications such as tissue engineering, can also be extended to wound dressing [18]. According to the literature, the application of NL for wound dressing is only restricted to lignin nanofibers, as Reesi et al. fabricated a gel that contains lignin nanofibers modified by arginine for wound-healing applications [19]. It was reported that the antioxidant and antibacterial activity of lignin increased nitric oxide content at the wound site. Additionally, nanosized lignin contains more hydroxyl and phenolic hydroxyl groups, which led to the electrostatic or hydrogen bond formation with different wound-healing agents and drugs [20]. Many researches have been explored the use of lignin in developing polyurethane foams [21,22], but there is no report on PU nanocomposite foams including NL. More particularly, although previous reports show the application of NL as additive in PU to develop advanced polymer nanocomposites [23,24], to the best of our knowledge, there is no evidence using such nanoparticles in PU nanocomposite foams. In addition, from the application perspective, only two pieces of research reported lignin being incorporated into commercial PU for tissue engineering and wound dressing purposes [25,26]. However, in the current contribution, for the first time, PU foam containing NL was synthesized for wound dressing applications.

Different types of approaches have been proposed to efficiently decrease bacterial load in wound dressings [27], among them propolis-coated wound dressing has been well known as an effective strategy due to its valuable medical and therapeutic features [28]. Compared with other natural antibacterial counterparts, e.g., green tea, cloves, and black tea extracts, propolis has shown to be a more active antimicrobial [29]. In addition, in competition with metal antimicrobials such as silver and copper, propolis exceeds their biocompatibility [30]. It has been also reported that this renewable antibacterial is antifungal, anti-inflammatory, antiviral, antitumor, antioxidant, and helps in pro-wound healing [31]. Kim et. al. fabricated biocompatible electrospun propolis/PU composite nanofibers using PU pellets (Skythane X595A-11) and it was suggested that the propolis could increase cell compatibility and antibacterial activity of composite nanofibers [32]. The fibrous composites were proposed as potential candidates in skin tissue engineering and wound dressing. In another work, Khodabakhshi et al. fabricated propolis/PU composite via commercial medical grade PU (Tecoflex EG-80A) using salt leaching/solvent casting method to prepare high porous PU foam coating with propolis and introduced a promising choice for wound dressing applications [33]. Regarding antibacterial efficacy of propolis, it significantly depends on both extraction method and solvent. Among various solvents, ethanol is more influential and common [34]. It is the reason that in the current study, ethanol was selected as an extraction solvent.

In the present study, soft porous polyurethane containing NL wound dressing was synthesized by solvent-free one-shot process, subsequently; the wound dressing was coated with the ethanol extract of propolis as a well-known antibacterial agent. The wound dressing was characterized from different points of view, including morphological, structural, mechanical, antibacterial, and biological (in vitro and in vivo) properties.

## 2. Materials and Methods

### 2.1. Materials

Kraft Lignin (Mn: 3000; Aldrich) was purchased from Sigma Aldrich (Shanghai, China). Polyethylene glycol (PEG, molecular weight of 400 g mol^−1^), PEG 600, glycerol, and 1, 6-diisocyanato-hexane (HDI) were supplied by Sigma-Aldrich (Shanghai, China) without further purification and used as received. The propolis was obtained from the Shahr-e Kord beehives, Iran. The cell culture materials including RPMI, Penicillin-streptomycin solution, fetal bovine serum (FBS), 0.05% trypsin/EDTA, and phosphate buffer saline (PBS) were bought from Bioidea (Shanghai, China). The MTT assay kit was purchased from Sigma-Aldrich (Shanghai, China). *Staphylococcus aureus* (ATCC 25,923) and *Escherichia coli* (ATCC 25922), L929 fibroblast cell line, Wistar rats were provided by Pasteur Institute of Tehran, Iran.

### 2.2. Methods

#### 2.2.1. Synthesis of Polyurethane Foam Containing NL

PU-NL foam was synthesized according to the following procedures. The preparation of NL was carried out via an ultrasonication treatment according to the referenced literature [35]. Briefly, an aqueous suspension of lignin was sonicated for 60 min and a homogeneous stable nanodispersion was obtained. Then, sonicated lignin was dried in mild conditions to be used in PU foam synthesis. PU-NL foam was synthesized with one shot and solvent-free process. Firstly, PEG 400, PEG 600, glycerol, and NL were premixed vigorously by a high-speed stirrer for 5 min in a plastic beaker to achieve homogeneous mixing. In the second stage, a calculated amount of HDI was added with a NCO/OH ratio at 1.2 and the mixture was heated gradually to 80 °C under stirring for 10 min. After that, water was added as a blowing agent and the mixture was stirred vigorously with a high-speed mixer at about 2000 rpm and was allowed to rise in free expansion at 80 °C. When the foam was no longer rising, it was placed in a vacuum oven at 70 °C for 24 h. It should be noticed that samples with various amounts of NL, ranging from 0 to 2.5 wt.%, were prepared; however, those without NL (as a comparison) and with 1 wt.% NL were selected for further characterization, considering the visual appearance of the sample as a wound dressing. Finally, the foam was cured at room temperature for 24 h before characterizations. After 3 days, samples were cut into specific shapes as per the test requirement, and the foam properties were then measured.

#### 2.2.2. Preparation of Propolis-Coated PU Foam Containing NL

The PU foams were coated by propolis as previously prepared in our group [36]. Briefly, the propolis was frozen at −20 for 24 h and then crushed in a blender. Propolis was dissolved in ethanol solution in 1:10 ratio (25 g of propolis in 250 mL of ethanol). The solution was maintained in a dark incubator at 37 °C for 14 days. After several filtering of the suspension using Whatman No. 4 filter papers, the solvent was removed by employing a rotary evaporator at 40 °C. The product was stored at 4 °C until further use. Finally, EEP-coated PU foams were prepared by soaking the PU foams into EEP, subsequently; they were dried at room temperature.

#### 2.2.3. Mechanical Properties

Tensile properties including Elongation at break and tensile strength of foamed samples were measured by a tensile test machine (Instron, model 5566, Ithaca, NY, USA) according to the ASTM D 882-02. At a speed of 10 mm/min and preload of 0.5 N the dumbbell shape samples (30 mm length and 5 mm width) were stretched to break. At least six samples were tested in each case.

#### 2.2.4. Fourier Transform Infrared Spectroscopy (FTIR)

The FTIR spectra of the PU-NL/EEP foams were collected by Fourier Transform Infrared Spectrometer (JASCO FT/IR-6100, accessory ATR PRO450-S), set on 4 cm^−1^ resolutions and 64 scans. Measurements were conducted at room temperature between 400 and 4000 cm^−1^ in transmission mode.

#### 2.2.5. Morphology and Density

The surface morphology and cell structure of propolis-coated polyurethane containing NL foams, PU-NL, and PU-NL/EEP was observed using scanning electron microscopy (SEM, TESCAN-Vega 3, Brno, Czech Republic) at 10 kV accelerating voltage. The samples were coated with gold nanoparticles using a vacuum gold sputter coater (Q150R-ES, Quorum Technologies, East Sussex, UK) before SEM analysis. The MATLAB and Image J software were used to calculate the average apparent porosity and diameter of the foams. For density determination, the length, width, and thickness of the samples were measured in millimeters and were weighed in grams. Then, density was measured at least 10 times with different samples as follows: density (g/cm^3^) = weight/ thickness × length × width.

#### 2.2.6. Water Absorption Test of PU-NL and PU-NL/EEP Foams

Water absorption was measured based on the foams’ weights after absorption in phosphate-buffered saline (PBS), at 37 °C for 24 h. Dry samples with dimensions of 1 cm × 1 cm were weighted as the Wd. Then, they were immersed in 5 mL of PBS and removed after being fully saturated. The weights of the swollen foams were recorded as the Ww after removing the excess moisture from their surfaces with filter papers. The water absorption ratio was calculated using the following formula: swelling ratio = [Ww − Wd]/Wd × 100%.

#### 2.2.7. Contact Angle Measurement

Water contact angle measurement was done by a contact angle measuring system (model GBX, Digidrop, Romans-sur-Isère, France) at room temperature. A drop of deionized water was deposited on the surface of the foams and images were converted by a computer to determine the wettability data.

#### 2.2.8. Particle Size Determination

Dynamic light scattering (DLS) is a technique used for sample particle sizing, typically in the sub-micron range. The mean particle size, particle size distribution, and mean zeta potential analysis of lignin nanoparticles were analyzed by DLS using a Zetasizer nano SZ-100 HORIBA-Japan Scientific nanoparticle analyzer at 25 °C. Before DLS analyses, handshaking was applied vigorously.

#### 2.2.9. Anti-Bacterial Test

For evaluation of anti-bacterial properties of the prepared samples (PU, PU-NL, PU-NL/WEP, and PU-NL/EEP), the zone of inhibition (ZOI) test was employed according to our previous studies [37]. In the case of anti-bacterial tests and to compare with EEP, water-extracted propolis (WEP) was also used for coating PU-NL, which was named PU-NL/WEP. *Staphylococcus aureus* (ATCC 25923) and *Escherichia coli* (ATCC 25922) were used as target pathogens. A total of 100 µL of bacteria containing medium after overnight incubation was spread over the nutrient agar medium by a glass L-rod. The samples were placed over the medium and incubated at standard conditions. After 24 h, the clearance zones around the discs were measured by a digital caliper. The test was done in triplicate.

#### 2.2.10. Cell Viability Assay

Colorimetric MTT assay was employed for assessment of the PU, PU-NL, and PU-NL/EEP samples with L929 Fibroblast. Circular samples with 1 cm diameter were cut from each film and sterilized, by exposure to UV light for 2 h and washing with 70% ethanol for 30 min. The sterilized samples were placed into the cell culture plates wells, which contained RPMI, and incubated for 1, 3, and 7 days at standard cell culture conditions (37 °C, 95% relative humidity, and 5% CO2). On each of the specified days, 100 μL of the incubated cell culture medium was transferred into a 96-well cell culture plate. Then, 5 × 103 L929 cells were suspended in standard cell culture media and added to each well. After 24 h incubation, the culture media was removed and wells were washed with PBS. Then, the cell viability was assessed according to the MTT kit manufacturer [38,39].

#### 2.2.11. Cell Adhesion

The PU-NL/EEP films were cut into circles with 1 cm diameter. Then, they were washed multiple times with PBS. To sterilize the films, they were exposed to UV light for 2 h, followed by a washing step using 70% ethanol for 30 min. The sterilized samples were put at the bottom of a 6-well culture plate and adequate cell culture media, which contained 1 × 105 L929 cells added to each well. The culture plate was incubated for 1 or 7 days at the standard cell culture condition. After three days, the samples were fixed with 4 mL of 2.5% glutaraldehyde. After 2 h incubation with the glutaraldehyde solution, it was discarded and the samples were washed using 50, 60, 70, 80, 90, and 100% ethanol. At last, the samples were lyophilized for 24 h, and cell morphology was assessed by employing SEM.

#### 2.2.12. Animal Skin Wound Model and Histological Analyzes

Male Wistar rats (200–250 g) were purchased from the Royan Institute of Isfahan, Iran. All animal experiments and protocols were approved by the ethical committee of the Isfahan University of Medical Science (IR.MUI.RESEARCH.REC.1397.103). At first, the rats were completely anesthetized by intraperitoneal injection of ketamine (100 mg/kg) and xylazine (10 mg/kg) solution. Then, the dorsal region skin of rats was completely shaved by an electric shaver and then disinfected by 70% alcohol. Subsequently, round excisions (11 mm in diameter) were made on the back of the rats using a punch biopsy. Subsequently, the animals were randomly divided into three groups (n = 8) including PU, PU-NL, and PU-NL/EEP. For all groups, the related wound dressings were applied precisely on the wound site and sterile cotton gauze was applied on the wound site. Each rat was kept in separate cages. The operation day was counted as the 1st day and the wound healing process was evaluated by measurement of wound diameters on the 5th and 10th day by a digital caliper. The percentage of the wound area closure was calculated by Equation (1) [2,3,4]. It should be mentioned that standardized humane endpoints based on the current guidelines for endpoints in animal studies were used [40,41,42].
(1)$$\mathrm{Wound}\;\mathrm{area}\;\mathrm{closure}\;\left(\%\right)=\frac{\mathrm{Wound}\;\mathrm{area}\;\mathrm{at}\;\mathrm{the}\;\mathrm{first}\;\mathrm{day}-\mathrm{Wound}\;\mathrm{area}\;\mathrm{at}\;\mathrm{day}\;\left(n\right)}{\mathrm{Wound}\;\mathrm{area}\;\mathrm{on}\;\mathrm{the}\;\mathrm{first}\;\mathrm{day}}\times 100$$

The animals were sacrificed by ketamine-xylazine mixture overdose on the 12th day after wound creation. Then, full-thickness skin excisions were made from the wound area (n = 8). The skin specimens were fixed in 10% formalin neutral buffer solution for 24 h. The fixed specimens were processed by an automatic tissue processor (Sakura, Japan). Then, 4 µm thickness serial sections from the paraffin-embedded blocks were prepared by a microtome (Leica Biosystems, Weztla, Germany). The sections were stained with Hematoxylin & Eosin (H&E) staining protocols [43] for subsequent histological evaluations under a digital light microscope (Olympus, Tokyo, Japan).

#### 2.2.13. Statistical Analysis

The quantitative data were displayed as mean ± standard deviation (SD). Statistical analyses for elucidating differences between groups were conducted using one-way analysis of variance (ANOVA) and Tukey’s HSD post-hoc test by SPSS software V.23. The difference was considered statistically significant if *p* < 0.05. (*: *p* ˂ 0.05).

## 3. Results and Discussion

### 3.1. Dynamic Light Scattering (DLS)

It is evident that obtaining nanosized lignin particles is possible by applying ultrasonic irradiation [35]. There are some approximations in the recorded data during the DLS analysis such as: the particles are homogeneous and spherical as well as optical properties of the sample and the environment are known. The mean particle size and Particle size distribution of NL were analyzed by DLS (Figure 1). Ignoring the second peak around 800 nm most probably related to nanoparticle agglomerations, it is evident that the light scattering profile showed an average particle size of 110 nm with a polydispersity index value of 0.9, where the mean particle size distribution is in the nano range. NL has a negative zeta potential for its negative charge at pH > 1 due to the ionization of its phenolic hydroxyl and hydroxyl groups [19]. According to the data, −0.3 mV resulted in zeta potential. The SEM images are in agreement with the DLS data as well. However, the ultrasonicated lignin particles were well into the nanometer domain.

### 3.2. Fourier-Transform Infrared Spectroscopy (FTIR) Analysis

FTIR spectra of NL, PU-NL, EEP, and PU-NL/EEP were characterized for vibrations of the functional groups, which infer molecular structure and chemical bonding, as reported in Figure 2. NL infrared absorption spectra show complex bands. This is due to the variety of vibration modes of chemical bonds present in this biopolymer structure. As shown, NL was confirmed by the main characteristic bands at 3400 cm^−1^ (O–H band vibration), 1601 cm^−1^ (aromatic ring vibration), 1515 cm^−1^ (aromatic skeletal vibration), C–H stretching of methyl or methylene groups at 2938 cm^−1^, C=O stretching at 1700 cm^−1^ and 1220 cm^−1^ (C-O stretching in phenol and ethers) [35].

The FTIR spectrum of PU-NL confirmed qualitatively the presence of urethane linkages. They are well represented by the characteristic -NH stretching vibration at 3333 cm^−1^ and the characteristic –CO vibration at1700 cm^−1^. No absorption band was observed at 2270 cm^−1^ assigned to isocyanate groups (–N=C=O) of HDI in the spectra of the PU-NL foam, which indicated that the isocyanate group was completely consumed with the formation of completed PU network containing NL. The absorption peak at 1100 cm^−1^ was the typical stretching vibration of C-O-C groups in the polyether-based PU foams. The absorption bands from 1605 to 1510 cm^−1^ correspond to the aromatic skeletal vibrations originating from lignin, and were also observed in the PU-NL spectrum, indicating that the main structure of lignin did not alter appreciably during the reaction [44]. Additionally, the chemical interactions between isocyanate and hydroxyl groups of NL were evidenced by decreasing a wider band at 3407cm^−1^ in NL and was merged with the OH bond at 3333 cm^−1^ in the PU-NL spectrum [22]. Furthermore, hydrogen bonds are formed between hydroxyl groups of NL and N-H (proton donors) and C=O (proton acceptors) in the urethane group (–NHCOO–) in PU-NL. From this result, it can be suggested that NL was well miscible in a matrix of PU foam at a molecular level [23,25].

In the EEP spectrum, the range of 3000–2800 cm^−1^ is the location of bands related to asymmetric and symmetric stretching vibrations of CH_2_ and CH_3_ groups. In the 3380 cm^−1^ peak, a very intense band is present, which is related to the O-H band at EEP. Also, this band was already expected, since EEP is prepared with ethanol as solvent. Also, the range of 3000–3500 cm^−1^ at the PU-NL/EEP spectrum was observed, which was wider than its counterpart peak at the PU-NL spectrum, which demonstrates appropriate coating of the polyurethane foam with propolis [36,45].

### 3.3. Mechanical Properties

Table 1 shows the mechanical properties of the prepared samples in tensile mode. As can be observed, compared to the pure PU foam, NL addition increased tensile strength, and elongation at break of the nanocomposite foam. Such improvement can be attributed to the nanoparticle addition, playing a role as reinforcement and its crosslinking effect of NL, likely both physical and chemical interaction [22]. On the other hand, PU-NL/EEP displayed a reduction in mechanical properties, yet more than PU, which can be due to the plasticizing effect of the EEP, as described earlier [36]. PU-NL/EEP meets the desirable tensile strength of wound dressing, as it was reported to be in the range of 0.8–18 MPa [46]. A similar trend was also observed for the tensile modulus of the samples, as an indicator of mechanical stiffness. In a previous report, Young’s modulus of the skin ranges from 10 kPa to 50 MPa, therefore, the fabricated dressings fit this limit [47].

### 3.4. Morphological Characterization

The SEM was used to evaluate the morphology and microstructure of the surface and cross-section of PU-NL foam and PU-NL/EEP wound dressing (Figure 3). This characteristic is important as the mechanical performance of the foam is highly influenced by the average cell size and the thickness of cell walls [48].

Generated CO2 gas as a by-product of one-step polymerization and foaming reaction is attributed to the specific porous structure of PU-NL foam’s outstanding pore interconnectivity and high porosity. As shown in Figure 3A,G, a comparison of 35× magnification micrographs for all foams shows very similar cellular structures and open-cell structures as expected. Also, the cross-section of foams was also observed in the form of homogeneous morphology as shown in Figure 3B,H. Distribution of NL is observed in 1500× and 8000× magnification micrographs of PU-NL (Figure 3D,E), which ranged from 45 to 80 nm as spherical nanoparticles and also some parts were slightly agglomerated. A range below 100 nm is evident for lignin as nanoparticles. Generally, lignin-based polyurethane foams produce smaller cell sizes and thicker cell walls than the pure PU, which might be due to the nucleating agent role of NL, explaining the higher density values and mechanical properties [48].

Furthermore, the influence of porous structure on the wound tissue infiltration in vitro as well as cell growth in vivo is reported [49] and the porous structure of polyurethane foams with a range from 50–350 μm can significantly benefit cellular neovascularization and infiltration [50]. The pore diameter of PU-NL was in the range of 150–200 μm (Figure 3F). Also, the EEP as the antibacterial agent was coated on the PU-NL foam surface and was penetrated to its pore walls according to morphological observations on the surface of the PU-NL/EEP wound dressing (Figure 3I).

### 3.5. Porosity and Density

The porosity for wound dressing can significantly impact the exudates’ absorption capacity, which can minimize the possibility of wound infection [51]. The apparent porosity of PU-NL and PU-NL/EEP foams was measured by image analysis, resulting in 92.3 and 87.9%, respectively. The addition of EEP to PU-NL foam caused a little difference in the porosity percentage of about 5% [52]. All foams have relatively uniform pore size and homogeneous morphology and are also highly porous, having a porosity of >85%. The average pore diameter of PU-NL and PU-NL/EEP was 110 μm. The densities of PU-NL and PU-NL/EEP foams were calculated as 0.21 g/cm^3^ and 0.28 g/cm^3^, respectively. In addition, EEP coating on PU-NL foam slightly increased the density for PU-NL/EEP wound dressing. It was also found that the higher the porosity, the lower the density.

### 3.6. Water Absorption Ability of PU-NL and PU-NL/EEP Foams

One of the determinative parameters for wound dressing fabrication is its capability to maintain moisture around the wound region as a result of outstanding water absorption related to the absorption of wound exudates [53,54]. PU foams are suitable for wound dressing due to their water absorption capacity [51]. The percentage of water absorption of PU-NL and PU-NL/EEP foams was calculated to be 267 and 242%, respectively. The water absorption of PU-NL/EEP foam was a little less than PU-NL, which can be related to the hydrophobicity nature of propolis due to the presence of fatty acids and terpenes at EEP and also decrease of porosity [55,56]. Although there was no significant difference in water absorption capacity, PU-NL/EEP foam can be used as a new wound dressing by the combination of antibacterial and endotoxin adsorption performance.

### 3.7. Contact Angle

The wettability of foams was evaluated by measuring the contact angle on the surface. Appropriate hydrophilicity is essential for the wound dressing’s surface due to contact with the wound area directly. An increase in hydrophilicity of PU foams makes it more biocompatible for skin tissue engineering and wound dressing [32]. Therefore, the hydrophilicity of the PU, PU-NL, and PU-NL/EEP foams was investigated and the water contact angle was determined as 98.3 ± 5.8°, 51.1 ± 4.9°, and 50.1 ± 2.1°, respectively (Figure 4). The incorporation of only 1 wt.% of NL into the PU foam reduced the contact angle by around 100%, most probably as a result of the free polar groups in NL [57]. It also revealed that the water contact angle of PU-NL/EEP wound dressing was decreased with the propolis coating due to different compounds of propolis that increase the surface hydrophilicity of dressing and also facilitate the cells’ attachment on the surface, which is crucial in wound healing [32].

### 3.8. Anti-Bacterial Activity

Infection at the wound site can deeply affect the healing process. In addition, wound infection can cause systemic complications like sepsis. Therefore, an ideal wound dressing should exhibit significant anti-bacterial properties [36]. One of the most well-known tests for the evaluation of wound dressing antimicrobial properties is the zone of inhibition test. In this study, the antibacterial activity of the prepared films (PU, PU-NL, PU-NL/WEP, and PU-NL/EEP) was assessed against *Staphylococcus aureus* and *Escherichia coli*, which are two of the most common wound infection-associated pathogens [58]. The PU and PU-NL films did not cause the formation of considerable inhibition zones for the used bacterial species, which indicates that these films’ anti-bacterial properties should be improved by incorporation of anti-bacterial agents (Table 2, Figure 5A). Therefore, the effect of incorporation of propolis water and the ethanolic extract was assessed on the anti-bacterial activity of the films. PU-NL/EEP and PU-NL/WEP exhibited significantly bigger zone of inhibition against both *S. aureus* and *E. coli* in comparison with PU and PU-NL films. Also, the PU-NL/EEP caused higher anti-bacterial properties compared with PU-NL/WEP. This observation is consistent with previous studies that demonstrated the advantages of EEP over WEP for anti-bacterial purposes [59,60]. EEP antibacterial activity can be attributed to the presence of phenolic compounds and flavanones with high antimicrobial effects [55]. PU-NL/EEP caused significantly lower antibacterial activity against *E. coli* in comparison with *S. aureus*. Previous studies have reported that structural difference in these two bacterial strains’ outer shells is the main reason for this observation. The cell wall of *E. coli* is covered by a thick membrane of lipopolysaccharide, while *S. aureus* has a single peptidoglycan layer. So, *E. coli* exhibits more resistance to hydrophobic antibacterial agents like EEP [60,61]. Taken together, as PU-NL/EEP exhibited more efficacy according to anti-bacterial properties compared with PU-NL/WEP, so this composite film was selected for further assessments in this study.

### 3.9. Cellular Biocompatibility

An ideal wound dressing should exhibit high biocompatibility with skin fibroblast [62]. As Figure 5B illustrates, the biocompatibility of PU, PU-NL, and PU-NL/EEP films was assessed with the skin normal fibroblast cells (L929 cell line). All the samples exhibited high biocompatibility with L929 cells at all time points. NL addition into PU, caused marginally higher viability, particularly in the 3rd and 7th day, as was reported elsewhere [63]. In addition, the highest cell viability was observed at the PU-NL/EEP films. Previous studies have reported the anti-proliferative effects of EEP on neoplastic cells. However, these effects of EEP were absent for normal cells including normal fibroblast in specific concentrations of EEP, which is consistent with our observations [64]. Also, many previous studies used EEP in their wound dressing structure and reported high biocompatibility with skin fibroblasts [55,56].

### 3.10. Fibroblast Adhesion to the PU-NL/EEP Wound Dressing

The surface of a wound dressing should prepare an appropriate matrix for skin fibroblast adhesion and cell supporting behavior [65]. As the PU-NL/EEP films were completely biocompatible according to the MTT assay, a reasonable cell attachment was expected. Figure 6 shows SEM images of L929 cells on the PU-NL/EEP films after 1 and 7 days. The PU-NL/EEP films had a suitable surface for cell adhesion, spreading, and growth, as Figure 6A,B represent. The number of cells was significantly higher in the surface of the PU-NL/EEP films on the 7th day in comparison with the 1st day. Also, the cells have attached and expanded their podocytes properly on the PU-NL/EEP films (Figure 6C), which demonstrate high interaction of cells adhesion molecules with the film structure.

### 3.11. In Vivo Wound Healing Assessment

The film’s effect as a wound dressing for accelerating the wound healing process was assessed in the Wistar rats’ full-thickness skin wound model. The wounds were treated with PU, PU-NL, and Pu-NL/EEP films in different groups (n = 8). No evidence of wound complications like necrosis or inflammation was observed in the treatment groups. As Figure 7A illustrates, the wound at the PU-NL/EEP group was approximately closed after 10 days from the operation. While wound at the other groups was still unhealed. As Figure 7B illustrates, at both time points (5th and 10th days), the PU-NL/EEP group exhibited a significantly (*p* < 0.05) higher wound closure rate compared with the control, PU, and PU-NL treated groups. The histological investigations approved these observations as the wounds treated with the PU-NL/EEP films exhibited more developed epidermis and dermis in comparison with other treatment groups [62]. Therefore, significantly higher wound healing activity was detected at PU-NL/EEP wound dressing-treated group in comparison with control, PU, and PU-NL groups.

## 4. Conclusions

The current contribution aimed to synthesize PU composite open-cell foam containing NL, coated by green antibacterial propolis, as a candidate for wound dressing applications. The open-cell structure of the foams was confirmed by SEM, an appropriate property for wound dressing. Both NL and propolis coating led to an increase in mechanical strength and hydrophilicity, compared with pure PU. Antibacterial activity of the samples was demonstrated against *Escherichia coli* and *Staphylococcus aureus*, which was significantly higher for PU-NL/EEP. In addition, in vitro cytocompatibility assessments showed that while all the foamed samples have high cell viability and cell adhesion, it is appropriately higher for PU-NL/EEP. Also, in vivo animal studies revealed that PU-NL/EEP promoted better skin wound healing.

## Figures and Tables

**Figure 1 polymers-13-03191-f001:**
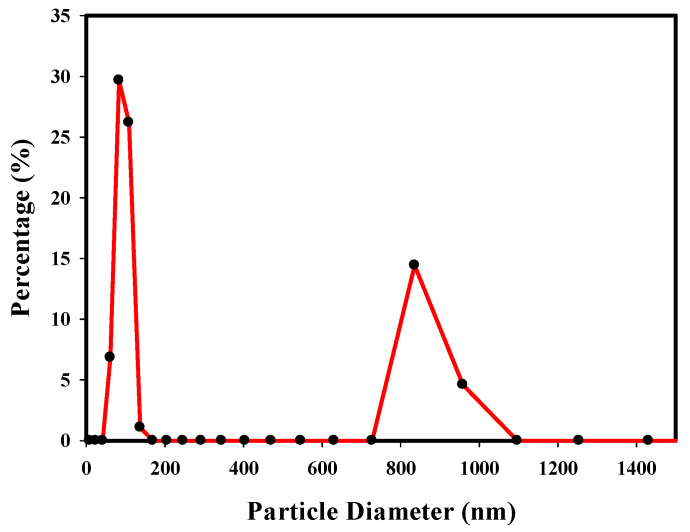
Particle Size Distribution of NL.

**Figure 2 polymers-13-03191-f002:**
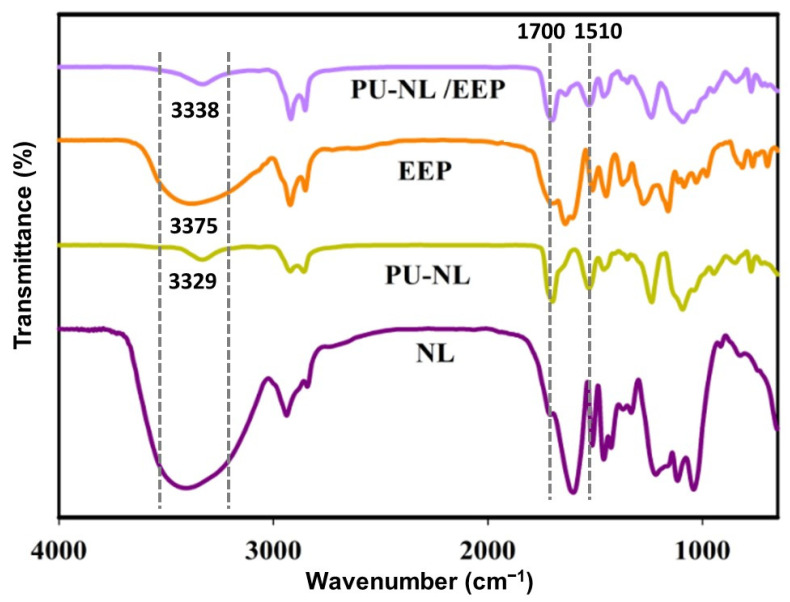
The FTIR spectra of NL, PU-NL, EEP, and PU-NL/EEP.

**Figure 3 polymers-13-03191-f003:**
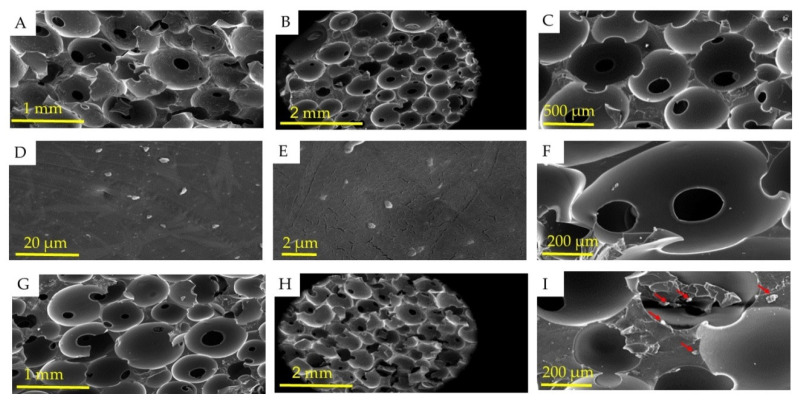
SEM images of PU-NL (**A**–**F**) and PU-NL/EEP (**G**–**I**) samples. The red arrows are displaying the propolis at the surface of the foam’s pore walls.

**Figure 4 polymers-13-03191-f004:**
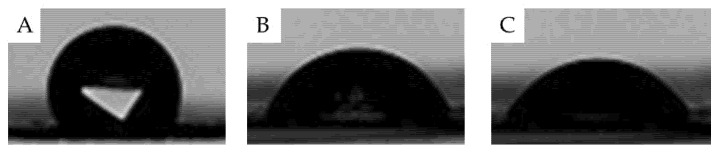
Contact angle images of the samples: (**A**) PU, (**B**) PU-NL, and (**C**) PU-NL/EEP.

**Figure 5 polymers-13-03191-f005:**
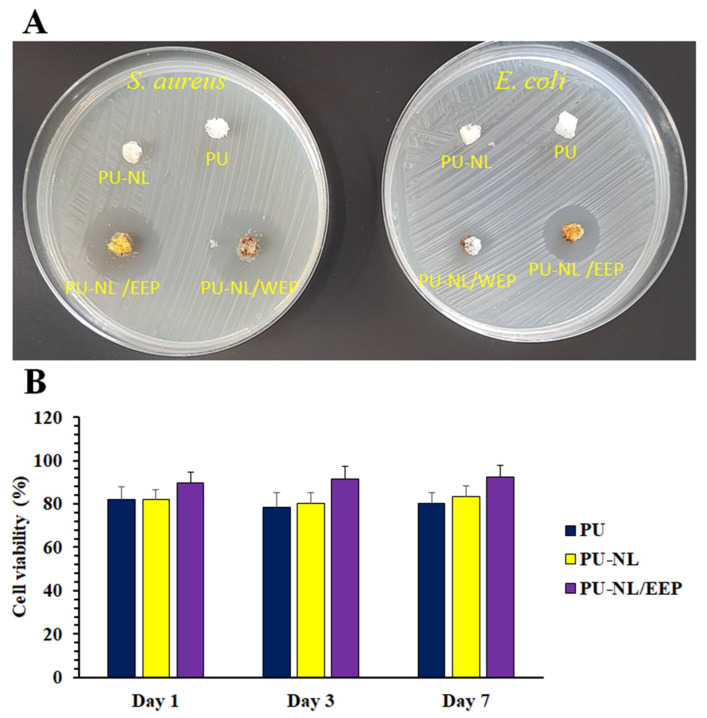
(**A**) The PU, PU-NL, PU-NL/EEP, and PU-NL/WEP films’ antibacterial activity against *Staphylococcus aureus* and *Escherichia coli* strains according to the zone of inhibition test. (**B**) Biocompatibility of the PU, PU-NL, PU-NL/EEP films with L929 skin normal fibroblast after different incubation times according to MTT assay analyzes.

**Figure 6 polymers-13-03191-f006:**
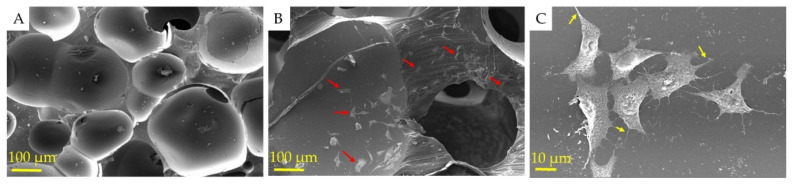
SEM images of the PU-Nl/EEP surface at the (**A**) 1st and (**B**) 7th days after L929 cells seeding. The red arrows indicate the L929 cells. (**C**) SEM images with higher magnification. Yellow arrows indicate L929 cells expanded podocytes.

**Figure 7 polymers-13-03191-f007:**
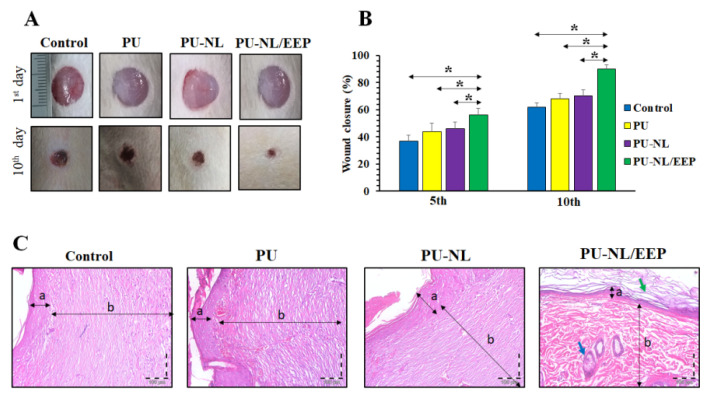
(**A**) Macroscopic appearances of the wounds at the 1st and 10th days post-operation at the PU, PU-NL, and PU-NL/EEP groups. (**B**) Histograms of the wound closure progression of the PU, PU-NL, and PU-NL/EEP groups. The data are expressed as mean ± standard deviation, (n = 8, *: *p* < 0.05, ns: not significant). (**C**) H&E stained sections of skin specimens from the wound site of the PU, PU-NL, and PU-NL/EEP groups. Two-head arrows indicated by a and b letters indicate the epidermis and dermis layers, respectively. Also, the green and blue arrows indicate the keratin layer and sebaceous gland, respectively.

**Table 1 polymers-13-03191-t001:** Mechanical properties of prepared samples.

Sample	Elongation at Break (%)	Tensile Strength (MPa)	Tensile Modulus (MPa)
PU	91 ± 3.5	0.75 ± 0.08	0.95 ± 0.15
PU-NL	96 ± 5.6	0.91 ± 0.1	1.25 ± 0.45
PU-NL/EEP	73 ± 3.9	0.82 ± 0.09	0.84 ± 0.19

**Table 2 polymers-13-03191-t002:** Antibacterial activity of the PU, PU-NL, PU-NL/WEP, and PU-NL/EEP films according to the zone of inhibition test.

Microorganism	PU	PU-NL	PU-NL/WEP	PU-NL/EEP
*E. coli*	0	0	4.3 ± 0.3	7.2 ± 0.4
*S. aureus*	0	2.1 ± 0.3	10.5 ± 0.7	11.2 ± 0.6

## Data Availability

The data used to support the findings of this study are available from the corresponding authors upon request.

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
