# Peer review of "Polyurethane-Nanolignin Composite Foam Coated with Propolis as a Platform for Wound Dressing: Synthesis and Characterization"

_polymers, 2021, doi:10.3390/polym13183191_

Round 1
Reviewer 1 Report
The authors reported a polyurethane-nanolignin composite foam coated with propolis as a platform for wound dressing. The research presented in this paper would be of interest to researchers working in polyurethane composite. The paper would be acceptable for publication after the following items have been addressed.
- In figure 1, why the particle diameter has two peaks?
- In Figure 2, I can’t see the difference among the spectra of PU-NL abd PU-NL/EEP. It is better to draw some guided lines and mark the important wavenumbers. Like this article (https://www.mdpi.com/2073-4360/12/9/1989)
- To evaluate the water absorption, it is better to present the percentage of water absorption at different time.
- In section 3.7, the pictures of water contact angle are supposed to be provided.
Author Response
The authors would like to appreciate the esteemed reviewer for his/her time and insightful comments. Please see the attached file.

Reviewer 2 Report
This study addresses the production of polyurethane foam composites, containing lignin that were further coated with natural antimicrobial propolis to improve antibacterial properties for wound dressing use. Moreover, fisico-chemical characterization, antibacterial properties, in-vitro and in-vivo tests were performed on those materials to test the potential for application. At this stage some revisions are required namely:
Introduction
Page 2, line 70 “…nanolignin (NL) which is sized from 1 to 1000 nm.” In the concept of nanoparticles, a nanoparticle is a small particle that ranges between 1 to 100 nanometres in size. Thus, we have to agree that 1000 nm it is on the micron scale. In the opinion of this reviewer this paragraph without a bibliographic reference, intends to provide a definition of nano-size particle that is not corrected and should be revised.
Materials and methods
Page 4, Mechanical tests: The followed methodology was according to the standard ASTM D 882-02, however only at least 3 specimens were tested. Typically, at least 5 specimens per condition are required to obtain an accurate value for mechanical properties. The author needs to test more samples.
Results and discussion
Page 6, DLS analysis: The author comment that in the figure 1 the lignin particle size is around 110 nm, however a considerable fraction ranges in a particle size higher than 800 nm that is in the micrometer range. Please comment on that.
Page 7, Mechanical properties: In the tensile properties of the developed composite materials it is missing the value of the modulus to infer about the stiffness of the materials. Thus, the discussion can be also improved taking in consideration the target application and the properties required for wound dressing.
Page 9, Figure 3: The scale bar of micrograph 3B and 3H, at lower magnification, should be the same (i.e. 1 mm) for comparison.
Page 9, Contact angle: In the paragraph line380 -382, the author states that after the addition of 1% of NL, the contact angle decreases almost 100º, however the contact angle for the PU foam is around 98.3º. Moreover, for the composite foams the contact angle values were around 50º. Please revise the sentence.
Author Response

(The authors gave the same response as above.)

Round 2
Reviewer 2 Report
The manuscript was revised according to the reviewer’s comments. The manuscript can be consider for publication in the present form.